# Aptamer against Aflatoxin B1 Obtained by SELEX and Applied in Detection

**DOI:** 10.3390/bios12100848

**Published:** 2022-10-09

**Authors:** Chung-Hsuan Yang, Ching-Hsiu Tsai

**Affiliations:** 1Graduate Institute of Biotechnology, National Chung Hsing University, Taichung 402, Taiwan; 2Advanced Plant Biotechnology Center, National Chung Hsing University, Taichung 402, Taiwan

**Keywords:** AFB1, SELEX, aptamer, aptasensor, qPCR

## Abstract

Aflatoxins, especially aflatoxin B1 (AFB1), are the most prevalent mycotoxins in nature. They contaminate various crops and cause global food and feed safety concerns. Therefore, a simple, rapid, sensitive, and specific AFB1 detection tool is urgently needed. Aptamers generated by SELEX technology can specifically bind the desired targets with high affinity. The broad range of targets expands the scope of applications for aptamers. We used an AFB1-immobilized magnetic nanoparticle for SELEX to select AFB1-specific aptamers. One aptamer, fl−2CS1, revealed a dissociation constant (*K*d = 2.5 μM) with AFB1 determined by isothermal titration calorimetry. Furthermore, no interaction was shown with other toxins (AFB2, AFG1, AFG2, OTA, and FB1). According to structural prediction and analysis, we identified a short version of the AFB1-specific aptamer, fl−2CS1/core, with a minimum length of 39-mer used in the AFB1-aptasensor system by real-time qPCR. The aptasensor showed a broad range of detection from 50 ppt to 50 ppb with an accuracy of 90% in the spiked peanut extract samples. With the application of the AFB1-aptasensor we have constructed, a wide range detection tool with high accuracy might be developed as a point-of-care testing tool in agriculture.

## 1. Introduction

Mycotoxins are toxic secondary metabolites produced by fungi associated with various crops and pose a severe threat to human and animal health when consumed [1]. Aflatoxins are highly toxic and the most prevalent natural substances among mycotoxins; they are produced mainly by *Aspergillus flavus* and *A. parasiticus*. There are 20 types of aflatoxins identified to date, of which six (B1, B2, G1, G2, M1, and M2) are the most predominant contaminants in food and agriculture products [2,3].

Aflatoxin B1 (AFB1) is classified as a group 1 carcinogen by International Agency for Research on Cancer. It causes liver damage, hepatocellular carcinoma, and deleterious health conditions [4]. The oral LD_50_ of AFB1 ranges from 0.3–18 mg/kg body weight, depending on the species [5]. Contamination with aflatoxin, especially AFB1, has thus become a main concern in global food safety.

High-performance liquid chromatography (HPLC) and liquid chromatography-mass spectrometry (LC-MS) are the two officially accepted analytical methods for AFB1 detection. However, these methods require expensive equipment, skilled staff, and sophisticated sample pre-treatment that limit them to laboratories. Enzyme-linked immunosorbent assay (ELISA) has been widely used as a rapid detection system in the field because of its simplicity of operation. However, time-consuming methods and high costs of the production of antibodies are major problems with ELISA. Furthermore, the instability of antibodies in different environmental conditions restricts their application [6]. Therefore, suitable alternatives for mycotoxin detection in food and agricultural products are urgently needed.

Aptamers are single-stranded oligonucleotides generated by the systematic evolution of ligands by exponential enrichment (SELEX) [7]. SELEX is an in vitro screening technique that involves incubating the randomized single-stranded oligonucleotide libraries containing 10^18^ sequences with the target. The selected sequences bound to the target are collected by physical or chemical methods and amplified by PCR for the next round of selection. After a few rounds of the same processes, sequences with high affinity and specificity can be identified [8,9]. The targets of aptamers include proteins, nucleic acids, toxins, cells, and even metal ions. The broad range of targets expands the scope of application for aptamers [10]. Aptamers applied for small-molecule detections have been reported, such as for mycotoxins [11], antibiotics [12], and pesticides [13]. The specificity of the selected aptamers with the target substances could derive from their 3-D structures. These are hydrophobic stacking, electrostatic complementarity, hydrogen bond formation, and structural complementarity [14,15]. The advantages of using oligonucleotides as aptamers are small size (20–60 bases), ease of synthesis and chemical modification, non-toxicity, and non-immunogenicity [16,17].

Aptamer-based sensors (aptasensors) have been used in various areas, including protein detection, environmental analysis, biomedical analysis, and diagnostics [18]. Recently, some aptamers were used to detect AFB1 with lateral flow assay [19], surface-enhanced Raman scattering [20], electrochemical detection [21], and real-time quantitative PCR (qPCR) [22]. Although these assays are highly sensitive and specific, they cannot be used for on-site detection because the instruments are not portable [23]. Furthermore, most of these aptasensors were derived from one AFB1 aptamer (International Publication Number WO2011020198A1; 50 mers with *K*d = 670 nM) that restricted its size and the cost of the production. Therefore, developing a set of rapid and portable detection platforms using the selected aptamer for AFB1 is urgently needed for detection on-site [24].

In this study, we used the SELEX technique to identify an aptamer that could specifically interact with AFB1. The aptamer was evaluated for its binding affinity and dissociation constant with isothermal calorimetry (ITC). We also shortened the length of the selected aptamer with a similar dissociation constant. The detection with spiked peanut extract samples was revealed by real-time qPCR and showed an acceptable detection range. Furthermore, we could combine the aptasensor system with a portable PCR machine and thus established a simple and rapid on-site aptasensor.

## 2. Materials and Methods

### 2.1. Materials

Aflatoxin B1, -B2, -G1, -G2, ochratoxin A (OTA), and fumonisin B1 (FB1) were purchased from Fermentek (Jerusalem, Israel). Carboxymethoxylamine hemihydrochloride (CMO), pyridine, dimethylformamide (DMF), potassium chloride, 1-(3-dimethylaminopropyl)-3-ethylcarbodiimide (EDC), Tween 20, and dimethyl sulfoxide (DMSO), glutaraldehyde, streptavidin, and KAPA SYBR FAST were purchased from Sigma-Aldrich (St. Louis, MO, USA). Sodium hydrogen phosphate, potassium dihydrogen phosphate, magnesium dichloride, calcium chloride, triton X-100, and sodium hydroxide were purchased from Merck (Darmstadt, Germany). Sulfo-NHS acetate and chloroform were purchased from Thermo Fisher Scientific (Waltham, MA, USA). TANBead U-118 was from Taiwan Advanced Nanotech (Taoyuan, Taiwan). Sodium chloride was from Union Chemical Works (Hsinchu, Taiwan). Tris was purchased from VWR International (Radnor, PA, USA). EDTA was purchased from Ameresco (Solon, OH, USA). Methanol was purchased from Echo Chemical (Miaoli, Taiwan). FavorPrep Nucarrier was from Favorgen Biotech (Ping-Tung, Taiwan). pGEM-T, PlasPrep kit was from Promega (Madison, WI, USA).

The sequence of the randomized single-stranded SELEX DNA library (5′-GAGAGGTCAGATG(N_30_)CCTATGCGTGCTAC-3′) was purchased from AllBio Science (Taichung, Taiwan). The sequences of the forward primer (5′-GAGAGGTCAGATG-3′), reverse primer (5′-GTAGCACGCATAGG-3′), biotinylated reverse primer (biotin- 5′-GTAGCACGCATAGG-3′), mid-13-base complementary DNA conjugated with qPCR template (5′-CTCTCGTAGCACGGGCACAGTGAAGTGAGACCACGCGGCCCATCGC CTCGCTGTCGGTGTG-3′), biotinylated aptamer (5′-CCATATGCGTGCTACGAGAGGTCAG ATAATGCACTATGG-3′), qPCR forward primer (5′-GGCACAGTGAAGTGAGACCACG-3′), and qPCR reverse primer (5′-CACACCGACAGCGAGGCGAT-3′) were purchased from MDBio (Taipei, Taiwan).

### 2.2. AFB1-MNP Preparation

To activate the functional group of AFB1, 1 mg AFB1 and 2 mg CMO were dissolved in 400 μL pyridine, kept in the dark, and shaken at 25 °C for 24 h. The solvent was evaporated with a speed vac concentrator (Savant, Thermo Fisher Scientific, Waltham, MA, USA). The residual pellet after removal of the pyridine was dissolved in 1 mL chloroform. The activated AFB1 was analyzed by thin-layer chromatography (TLC) run in parallel with the AFB1 standard. The activated AFB1 was then dissolved in 400 μL DMF after removal of chloroform with the speed vac concentrator. To prepare the AFB1 beads, 1 mL EDC (5 mg/mL) was added drop by drop to the activated AFB1 and incubated at 37 °C for 5 min. The aminated magnetic nanoparticles (MNPs) were added to the solution, kept in the dark, and incubated at 37 °C for 3 h. Sulfo-NHS-acetate (10 mg/mL) about 1 mL was then added and incubated for 2 h. AFB1-MNPs were collected with the help of a magnet and dissolved in PBS-T buffer at 4 °C for further use [25]. The recovery efficiency of activated AFB1 was quantified by enzyme-linked immunosorbent assay (ELISA) (Vaccigen Biomedical Technology, Taipei).

### 2.3. AFB1 Aptamer Selection

Approximately 1 μM of the randomized single-stranded DNAs (ssDNAs) (the first cycle of SELEX) in 200 μL binding buffer (100 mM NaCl, 20 mM Tris-HCl pH 7.0, 2 mM MgCl_2_, 5 mM KCl, 1 mM CaCl_2_, 0.02 % Tween 20) was incubated in a dry bath at 95 °C for 5 min and transferred to the ice bath for 5 min. We prepared 2 μL AFB1-MNPs (~50 ng), transferred them to a microfuge tube, and discarded the supernatant with the help of a magnet. The ssDNAs on the ice bath were added to the AFB1-MNPs and incubated at 37 °C for 30 min. After supernatant removal, the AFB1-MNPs were washed five times with 200 μL binding buffer and suspended in 20 μL ddH_2_O. The AFB1-interacted ssDNA was eluted after incubation on a dry bath at 95 °C for 5 min and then on an ice bath for 5 min. The eluent was used as the PCR template.

PCR was set in a total of 20 μL reaction containing 4 μL PCR template (the eluent from AFB1-MNPs), 3 μL forward primer (10 μM), 3 μL biotin-labeled reverse primer (10 μM), 1 μL MgCl_2_ (25 mM), 1 μL dNTP (2.5 mM), 2 μL 10X PCR buffer (500 mM KCl, 100 mM Tris-HCl pH 8.8), 5 μL ddH_2_O, and 1 μL Taq DNA polymerase (New England Biolabs, Ipswich, MA, USA). The reaction condition was set at 95 °C for 5 min, then 30 cycles of 95 °C for 30 s, 50 °C for 30 s, and 72 °C for 30 s, followed by 72 °C for 5 min. The PCR products were analyzed by polyacrylamide gel electrophoresis (PAGE).

Because the double-stranded DNA (dsDNA) is labeled with biotin by the reverse primer during PCR, dsDNA was then incubated with Dynabeads MyOne Streptavidin C1 (Thermo Fisher Scientific) for 30 min. After washing, the non-biotinylated strand was eluted by the addition of 50 μL of various concentrations (10, 20, 75, or 150 mM) of NaOH [26]. The eluent ssDNA was precipitated with the addition of 10 μL FavorPrep Nucarrier (Favorgen Biotech Corp., Pingtung, Taiwan) and 110 μL 100% ethanol, then centrifuged at 13,000 rpm (Heraeus Pico17, Thermo Fisher Scientific) at 4 °C for 5 min. After removing the supernatant, the DNA pellet was dissolved in 10 μL ddH_2_O. The ssDNA product was analyzed by PAGE. The concentration of ssDNA was estimated and diluted to 10 nM in a 200-μL binding buffer for the next SELEX cycle.

### 2.4. AFB1 Aptamer Cloning

After the last round of SELEX, the PCR products were eluted from a polyacrylamide gel. The dsDNA end product was cloned into a pGEM-T vector (Promega, Madison, WI, USA). The condition for the ligation was set as 0.5 μL pGEM-T vector (25 nM), 1.5 μL dsDNA insert (50 nM), 5 μL 2× ligation buffer, 0.25 μL T4 DNA ligase (3 units/μL) (Promega), and 2.75 μL ddH_2_O. The mixture was incubated at 16 °C for 2 h. Then the ligated product was transformed into *E. coli* DH10B. The positive clones (with blue/white selection) were cultured, and the plasmids were isolated by using the PlasPrep Kit (Promega). The insert-containing plasmids were sequenced with an ABI 3730XL DNA Analyzer served by G-TeC Genomic Technology Core (Academia Sinica, Taiwan).

### 2.5. Binding Affinity and Specificity of AFB1 Aptamer

To examine the binding affinity and specificity of the selected aptamer, we used isothermal titration calorimetry (ITC) MicroCal iTC200 (Malvern Panalytical, Malvern, UK) [27]. AFB1, AFB2, AFG1, AFG2, OTA, and FB1 were dissolved in DMSO and added to the buffer. The thermal equilibration was set at 25 °C with an initial 60-s delay step with 125 μM AFB1, AFB2, AFG1, AFG2, OTA, or FB1 in the syringe and 5 μM selected ssDNA aptamer in the sample cell dissolved in buffer (0.1 M Tris-HCl pH 7.4 in 1% DMSO). The binding experiment involved using a 0.4-μL injection at the first injection, followed by a 17-consecutive 2.1-μL injection every 60 s. The reference power setting was 5 mcal/s, and the syringe stirring speed setting 800 rpm. The control was set up under the same conditions as in the binding experiment by injecting the toxins without the addition of DNA samples. Calorimeter software was used to analyze the data.

### 2.6. Structural Prediction and Docking Study of Selected AFB1 Aptamer

To reveal the possible interaction between AFB1 and the selected aptamers, the secondary structure of the aptamer and its derivatives was predicted by using the mfold web server (http://unafold.org; accessed on 19 August 2022) under the condition as linear at 37 °C with the ionic concentration of Na^+^ and Mg^2+^ setting at 100 and 2 mM, respectively [28]. The tertiary structure of the aptamer was generated with the equivalent RNA models by the 3dRNA web server (http://biophy.hust.edu.cn/3dRNA; accessed on 26 April 2022) [29]. The 3D RNA models were converted into 3D DNA structures by using Discovery Studio and PyMol software [30,31]. In the docking study, Vina Wizard of PyRx software was used to predict the docking site of AFB1 and its specific aptamer.

### 2.7. Competitive AFB1 Aptasensor with Spiked Sample

The selected AFB1 aptamer with the best binding affinity was applied to the aptasensor system as described [22]. Real-time qPCR tubes were treated with 300 μL of 1% glutaraldehyde at 37 °C for 15 h and washed three times with ddH_2_O. Approximately 300 μL of streptavidin dissolved in ddH_2_O (1 μg/mL) was added and incubated at 37 °C for 1.5 h and washed two times with PBST (6.7 mM Na_2_HPO_4_, 3.3 mM NaH_2_PO_4_, 0.05% Tween-20, pH 7.2). Then 10 μL of 5′-biotinylated aptamer and its complementary DNA conjugated with qPCR template dissolved in hybridization buffer (750 mM NaCl, 75 mM sodium citrate pH 8.0) were added to the tubes and incubated on ice for 30 min. After washing three times with hybridization buffer, 0, 50, 500, 5000, 50,000 ppt AFB1 dissolved in buffer (10 mM Tris-HCl pH 7.0, 120 mM NaCl, 5 mM KCl, 20 mM CaCl_2_, 1% DMSO) was added to each tube and incubated on ice for 5 min; this step was repeated three times. Real-time qPCR was applied in a 20 μL-reaction comprising 10 μL KAPA SYBR FAST (Kapa Biosystems, Wilmington, MA, USA), 0.4 μL qPCR forward primer, 0.4 μL qPCR reverse primer, and 9.2 μL ddH_2_O. The real-time qPCR was performed in a thermocycler (TOptical Gradient 96, Biometra) and set as the initial denaturation for 3 min at 95 °C, followed by 40 cycles of denaturation for 3 s at 95 °C and annealing/extension for 20 s at 60 °C. Fluorescence measurements were taken after each annealing step. A melting curve analysis was performed from 60 °C to 95 °C.

Organic peanuts purchased from the local market were used in the competitive AFB1 aptasensor system as spiked samples. Approximately 100 g peanuts was ground, dissolved in 100 mL of 80% methanol, and shaken for 10 min. The supernatant was collected after 5 min of 3000 rpm centrifugation and stored at 4 °C for further use. Different concentrations of AFB1 dissolved in DMSO were added to peanut extract that was applied on spiked sample test with the same method mentioned above but replacing the Tris buffer with peanut extract during incubation.

## 3. Results and Discussion

### 3.1. The Preparation of AFB1-MNPs

Because AFB1 does not contain any active functional group, the carbonyl group of cyclopentanone (Ring 1) in AFB1 (Figure 1A) was chosen to be converted to a carboxyl group (Figure 1B). The conversion result was examined on a TLC plate. AFB1 moved to the top of the plate (lane S, Figure 1C), whereas the converted product, AFB1-oxime, was retained at the starting point (lane A, Figure 1C). The converted form AFB1-oxime did not interfere with the movement of AFB1 when mixed (lane M, Figure 1C). The reaction mixture of AFB1 and AFB1-oxime recovered from the conversion was then treated with EDC (Figure 1B). The activated AFB1-oxime could form an unstable intermediate and react with the amino groups of the aminated magnetic nanoparticles (TANBead U-118) in the presence of sulfo-NHS-acetate (Figure 1B). Subsequently, AFB1 was immobilized onto the magnetic beads to form AFB1-MNPs. The optimization results indicated that the activated AFB1 (~145 μg/200 μL reaction) incubated with 25 μg MNPs had the best yield, approximately 43% AFB1-MNP after ELISA (Appendix A). Too much MNP added in the reaction tended to be aggregated, which might interfere with the conjugation process.

### 3.2. Aptamer Selection with SELEX

The ssDNA pool consists of a 30-nt central randomized sequence flanked with 13 and 14 nt of the known sequence (the forward and reverse primer annealing site) at its 5′- and 3′-end, respectively. We optimized the PCR condition to obtain a consistent and best yield with the SELEX process to acquire correct and specific products. After optimization, the best reaction setting was annealing at 58 °C with 30 cycles (Appendix A).

We used the biotinylated primer for PCR followed by incubation with streptavidin beads (Appendix A). After removing the unbound products, the beads were treated with alkaline for optimizing the elution condition (Appendix A). The non-biotinylated ssDNA (the selected aptamer) could be recovered from treatment with 20 mM but not a lower concentration of NaOH (Appendix A).

In SELEX, the most critical step is converting the dsDNA PCR products into ssDNA for functional analysis. After a few rounds of the 9-cycle-positive selection (Appendix A), we sequenced 30 clones derived from the 9th SELEX product. According to the sequencing data of these clones, the aptamers were synthesized (Table 1). The aptamers were examined for their binding affinity with AFB1 by using isothermal titration calorimetry (ITC). Even though one of the aptamers appeared in eight clones, we could not detect the binding value without (Table 1, *K*d) or with the flanking sequence (Table 1, *K*d/fl). The results suggested that the aptamer might be selected against beads (MNPs) instead of AFB1 on the MNPs. Therefore, we tried one round of counter selection (negative selection) against NH2-MNPs (Appendix A) after the first round of positive selection and ended with positive selection until the 9th round of SELEX. Four of the 10 clones sequenced showed the same sequence, designated 2CS1 (Table 1). These aptamers were synthesized and examined by ITC. 2CS1 showed no interaction with AFB1, but 2CS1 containing the flanking sequence (fl−2CS1) had a *K*d of about 2.5 μM (Figure 2A). Therefore, the counter selection is critical to remove the nonspecific interaction against the beads. Furthermore, the flanking sequence used for PCR priming was also involved in the selection process. According to the sequence of fl−2CS1, structural prediction with the mfold web server revealed two alternate structures with changes of Gibbs free energy (Δ*G*) of about −8.26 and −7.56 kcal/mol (Figure 2B).

### 3.3. The Specificity Test of the Selecte Aptamer fl−2CS1

To inspect whether fl−2CS1 is specific to AFB1, we examined the binding activity of fl−2CS1 with other toxins. The results revealed no binding activity of fl−2CS1 with different types of aflatoxin under the same condition on ITC (Figure 3). The specificity of the aptamers against their targets mostly relies on the tertiary structure of the aptamers [32,33]. For OTA and FB1, the non−binding may be attributed to their larger size. For AFB2, AFG1 and AFG2, although their structures are similar to that of AFB1, the minute difference in the ring structure (Ring 5 between AFB1 and AFB2; Ring 1 between AFB1 and AFG1/2) could significantly interfere with the binding. Thus, the binding between fl−2CS1 and AFB1 could be size-exclusive and structure-dependent.

### 3.4. Structural Optimization of the Selected Aptamer

Because fl−2CS1 has two alternate structures with similar ΔG, we wanted to know which was responsible for binding AFB1. We constructed two fl−2CS1 mutants (T26A and C28T) by a structural prediction that could fix either one of the structures and examined by ITC. fl−2CS1/T26A (Figure 4A) and fl−2CS1/C28T (Figure 4B) had a similar binding affinity to AFB1, with *K*d 2.6 and 4.0 μM, respectively. We also constructed two other mutants to examine the significance of the single-stranded regions. One had four additional nucleotides at the 3′-end and two nucleotide substitutions that could reduce the size of the 5′-end single-strand portion (fl−2CS1/3′-addition, Figure 4C). The other had the removal of the 3-nt bugle near the 3′-end (fl−2CS1/Δ3′bulge, Figure 4D). ITC revealed that these single-stranded regions are critical for AFB1 binding. Overall, these results suggested that the binding pocket responsible for the AFB1 binding is the common region of fl−2CS1/T26A and fl-2CS1/C28T that includes the flexible 5′- and 3′-end and the 3-nt bulge near the 3′-end. Accordingly, we constructed a short version of the aptamer with the 39-mer containing all the critical regions (fl−2CS1/core, Figure 4E). ITC of the fl−2CS1/core revealed a similar binding affinity (*K*d = 4.4 μM) to that of fl−2CS1 (Figure 2A and Figure 4E).

To reveal the possible interaction mode of the fl−2CS1/core with AFB1, we generated the 3D structure model and fed AFB1 to the model by the docking program to predict the possible interaction sites. Three regions in fl−2CS1/core could accommodate nine possible interactions: three are in Region 1, one is in Region 2, and five are in Region 3 (Figure 5). The Δ*G* of these interactions ranged from −8.2 to −7.1 kcal/mol (Appendix A). The binding site in Region 2 was the most stable and conserved among all derivatives of fl-2CS1.

### 3.5. AFB1 Aaptasensor and Its Application on Spiked Samples

From the ITC measurement, the fl−2CS1/core could be used as the AFB1 aptasensor. The sensor design was set to use the 5′-biotinylated fl−2CS1/core to apply for competitive binding to AFB1 (Figure 6). The biotinylated aptamer interacted with the streptavidin-coated PCR tube and is complementary to the DNA probe. In the presence of AFB1, the aptamer targets the AFB1 and releases the DNA probe. Therefore, the more AFB1 present in the sample, the more DNA probe is released and the less DNA probe is retained on the aptamer. The retained probe on the aptamer could be quantified by qPCR (Figure 6B). The Ct value was increased from 14.12 to 18.06 when AFB1 was present from 0 to 50 ppb. This result supports the feasibility of the competitive aptasensor system. To confirm that our aptasensor system could work in reality, we used the spiked samples (the peanut extract) for the test. The aptasensor system could detect the AFB1 ranging from 50 ppt to 50 ppb with a recovery ratio from 96.4 to 90.2% (Table 2). The minimum accuracy of the system, taking into account 50 ppt to 50 ppb in the spiked samples, is about 90%. In the comparison of the aptasensor developed, our aptasensor is not the most sensitive (0.6 ppt with surface-enhanced Raman scattering system; SERS) and does not cover the widest range (1 ppt to 100 ppb with SERS or Fluoresce) (Table 3), but fulfills the detection of food safety allowance limit for direct consumption (2 ppb for the European Unit and 20 ppb for USA) [34].

These results suggest that our aptamer-based AFB1 biosensor shows strong potential to maintain accuracy and is feasible to module into a point-of-care testing (POCT) application. POCT devices must be portable with a short detection time and be easy to handle in the field. In our current system, the total detection time from the sample extraction to the end of qPCR is about 1.5 h. The aptasensor system we have developed is under construction for adoption as a handheld real-time PCR device in the near future [35].

## 4. Conclusions

We have successfully selected an aptamer against AFB1 by SELEX technology. SELEX is a powerful tool that can select a nucleotide combination against the desired target. In our first attempt of SELEX, the outcome aptamers were mainly against magnetic nanoparticles. Once using the counter selection, we could obtain the aptamers against AFB1. Of note, the flanking nucleotides used for PCR priming sites were also involved in the selection process. The selected aptamers without the flanking nucleotides revealed no interaction with AFB1. According to the structural analysis, we have identified the shortest aptamer, 39-mer, for the system to detect AFB1. The detection accuracy can reach 90% from 50 ppt to 50 ppb in the spiked samples. The accuracy of our aptamer-AFB1 sensor is feasible for design of a POCT system. In our current system, the total detection time from the sample extraction to the end of qPCR is about 1.5 h. In the future, we can optimize all the conditions to shorten the time to less than 1 h for the entire process.

## Figures and Tables

**Figure 1 biosensors-12-00848-f001:**
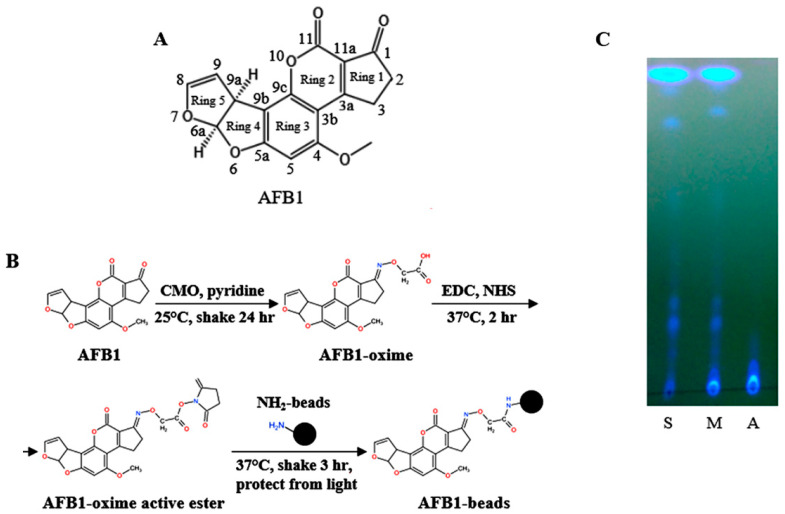
Preparation of the AFB1-MNPs. (**A**) Structure of AFB1. (**B**) Reaction of the preparation of AFB1-linked magnetic nanoparticles (MNPs). (**C**) Thin-layer chromatography analysis of the activated AFB1. S: AFB1 (5 μL), A: activated AFB1 (5 μL), M: AFB1 and activated AFB1 mixture (10 μL).

**Figure 2 biosensors-12-00848-f002:**
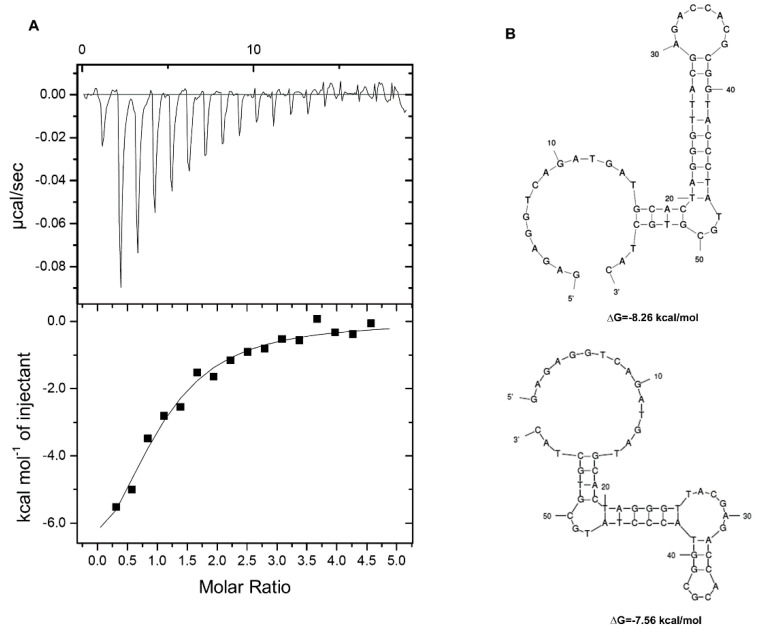
Binding affinity of the aptamer fl−2CS1 with AFB1 determined by isothermal titration calorimetry (ITC). (**A**) ITC measurement of the selected aptamer fl−2CS1 with AFB1. (**B**) Structural prediction of the fl−2CS1 and the changes of Gibbs free energy.

**Figure 3 biosensors-12-00848-f003:**
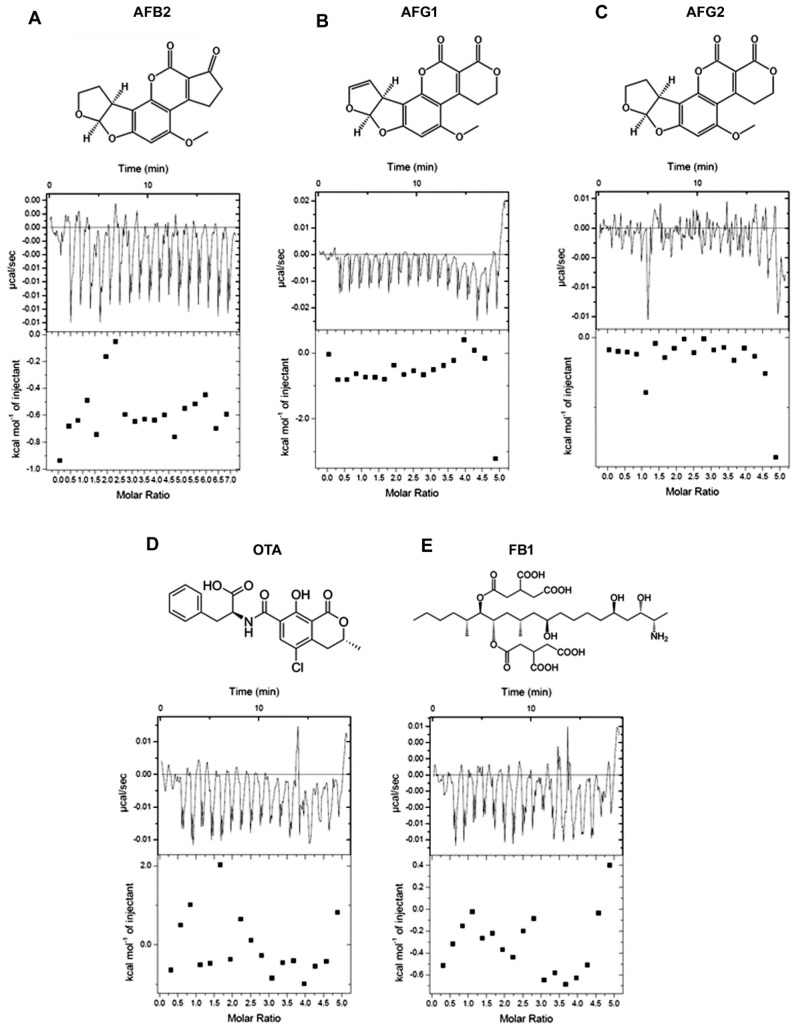
Binding affinity of aptamer fl−2CS1 with (**A**) AFB2, (**B**) AFG1, (**C**) AFG2, (**D**) ochratoxin (OTA), and (**E**) fumonisin B1 (FB1) determined by ITC.

**Figure 4 biosensors-12-00848-f004:**
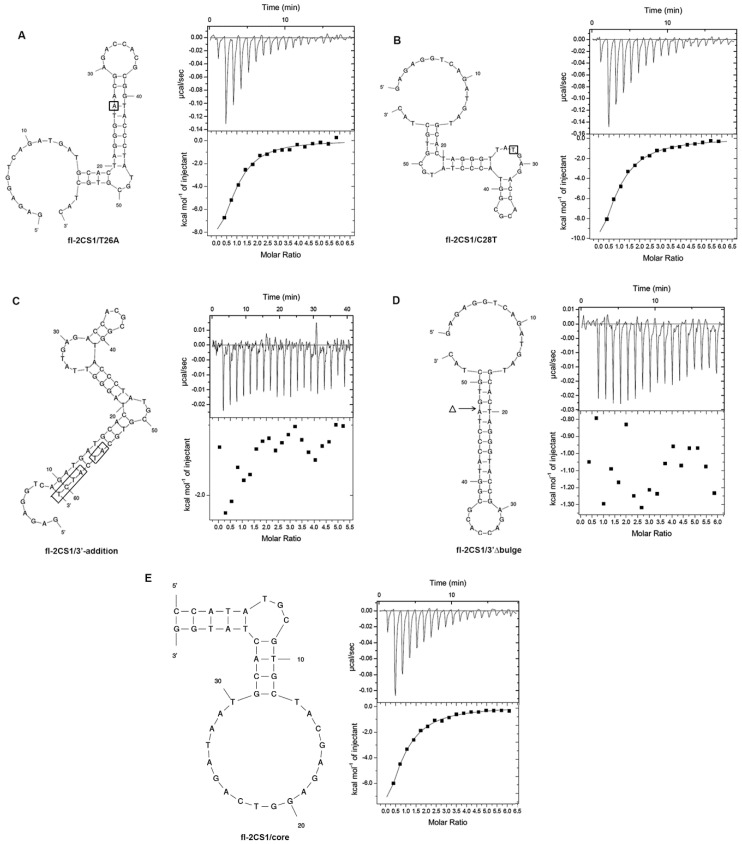
Binding affinity of the fl−2CS1-derivatives (**A**) fl−2CS1/T26A, (**B**) fl-2CS1/C28T, (**C**) fl−2CS1/3′-addition, (**D**) fl−2CS1/Δbulge, and (**E**) fl−2CS1/core with AFB1 determined by ITC. The mutation sites are boxed or indicted.

**Figure 5 biosensors-12-00848-f005:**
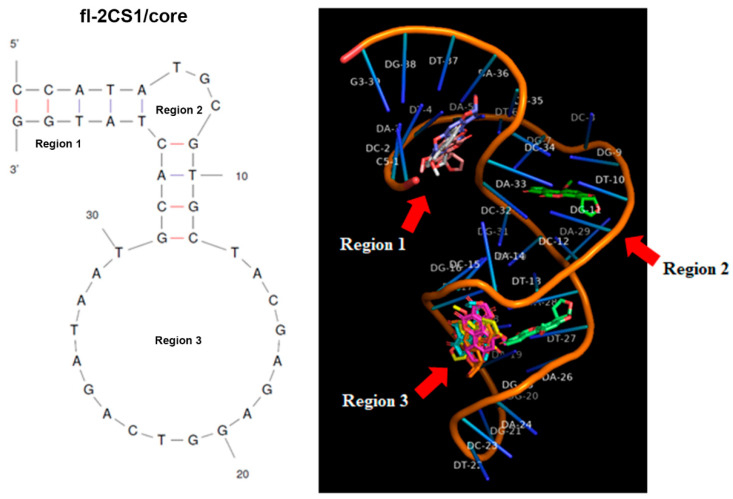
3D structural model of fl−2CS1/core and its possible docking sites with AFB1.

**Figure 6 biosensors-12-00848-f006:**
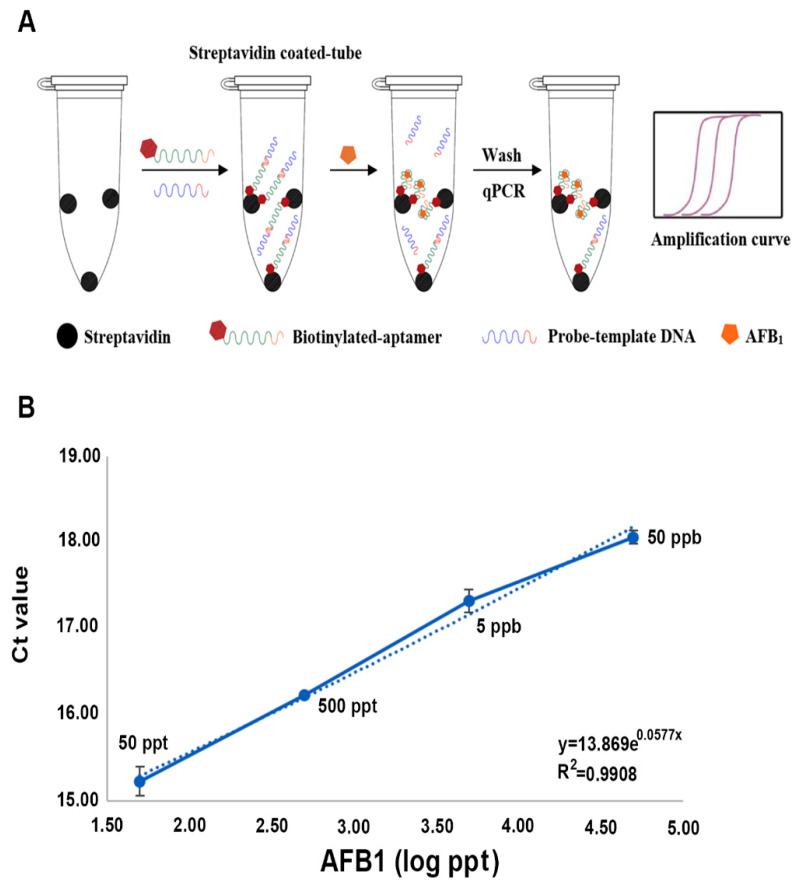
The competitive AFB1 aptasensor system. (**A**) Schematic illustration of the working system of the aptasensor. (**B**) Dynamic range of real-time PCR in determining the amount of AFB1.

**Table 1 biosensors-12-00848-t001:** The sequence and characteristics of the aptamers in this study.

Aptamers ^a^	Sequence ^b^	Length (nt) ^c^	*K*d (μM) ^d^	*K*d/fl (μM) ^e^	Numbers ^f^
S1-1	GTCAACACCGCACACATATATATGTTGGG	29	N	N	8
S1-2	GGCAAGTGCACCCGCATAGTTTTCGCCCC	29	N	N	4
S1-4	GTCACCGACCTGCCCGCATCGGTTGCTCC	29	N	N	1
S1-10	GTGCGGGTGGCCCGCACGCATTACGCGTTC	30	N	N	1
S1-14	GCCAGGCGGGGTGTTGAGTGCCGCCATATG	30	N	N	1
S1-16	GTACGCAGGATCACGCATTCACTATCGCTC	30	N	N	1
S1-22	CGTTAGGGAGGGAGTATCACCACGCGCTAC	30	N	N	3
S1-23	GTGCATGAACTGACCACGCGGTCCTAGGTC	30	N	N	1
S2-4	GTGTTGGCCTGGGACCATACCACGCGCTAC	30	-	N	2
S2-8	TCAACACCGCACACATATATATGTTGGG	28	-	N	1
S2-17	GGGTACATCGACCGCACGTATATGTTAC	28	-	N	1
S3-3	GCCCCCACGCTCTTGAGAGGACACGGCCCA	30	-	N	1
S3-4	GTCCGTTAGTTCGTTATCCCGGGGTTCCCA	30	-	N	1
S3-15	CTATAACGGCGTATGACCGTGTGCACCCCA	30	-	N	1
S5-8	GGCACAGGCTAAAAATTGGACGCGTTCCCA	30	-	N	1
S5-9	GTAATGTCTGATGGATCCTCCATCGGCCCA	30	-	N	1
S5-14	TCCATGCCGCCGACCAGTTTCACCACCCCA	30	-	N	1
Selection with one round of counter selection against NH_2_-MNPs					
1CS4	GTCCAAGTGCAATGGAACCACGCGGCTGTG	30	-	N	1
1CS7	GTGCGGAGCGAGCTGACCACGCGGCAGGTG	30	-	N	1
1CS9	GGTGCAGATCTCGATCTGACCACGCGGTCC	30	-	N	1
2CS1	ATGCACTAGGGTTACGAGACCACGCGGTAC	30	N	2.5	4
2CS8	CCCTGGCCGCCCCGCATAGGTGTGGTC	27	-	N	1
2CS9	GTGCACTGACCGCCCGCATAGCATGGTGTG	30	-	N	1
2CS12	GGCACATATGACCCGCATAGGCAGTTGTC	29	-	N	1

^a^ Name of each individual sequence clones. ^b^ Sequence of the aptamer. ^c^ Length of each aptamer selected. ^d^
*K*d of the selected aptamer without the flanking known sequence; N: non-detectable from ITC, -: not done. ^e^
*K*d of the selected aptamer with the flanking known sequence. ^f^ The total number of clones with the same sequence was selected.

**Table 2 biosensors-12-00848-t002:** Detection of AFB1 in the spiked samples.

AFB1 (ppt) ^a^	Average Detected (ppt) ^b^	Recovery Ratio (%)	RSD (%) ^c^
50	47.727	95.5 ± 2.5	2.63
500	481.892	96.4 ± 4.3	4.46
5000	4570.822	91.4 ± 5.9	6.45
50,000	45,098.527	90.2 ± 0.8	0.91

^a^ Input amount of AFB1 added to the spiked samples. ^b^ The average amount of AFB1 detected by the aptasensor with three independent experiments of the three repeats in each experiment. ^c^ RSD is the relative standard deviation.

**Table 3 biosensors-12-00848-t003:** Comparison of the developed AFB1 aptasensors.

Methods	LOD (ppb)	Linear Range (ppb)	References
SERS ^a^	6 × 10^−4^	0.001–100	[36]
PEC ^b^	2 × 10^−3^	0.01–100	[37]
AIE-aptamer-GO system ^c^	0.25	0.1–150	[38]
FRET ^d^	6.7 × 10^−3^	0.01–100	[39]
Fluorescence	8.9 × 10^−4^	0.001–100	[40]
Real-time qPCR	9 × 10^−3^	0.05–50	This work

^a^ SERS: surface-enhanced Raman scattering. ^b^ PEC: photoelectrochemical. ^c^ AIE-aptamer-GO system: aggregation-induced emission-aptamer-graphene oxide system. ^d^ FRET: fluorescence resonance energy transfer.

## Data Availability

The data supporting the findings of this study are available from the corresponding author (CHT) upon request.

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
