# Peer review of "Aptamer against Aflatoxin B1 Obtained by SELEX and Applied in Detection"

_biosensors, 2022, doi:10.3390/bios12100848_

Round 1

Reviewer 1 Report

 In this paper, an aptamer against AFB1 was selected through SELEX and the aptamer was cut off based on structural prediction and analysis. Then an AFB1-aptasensor system was built to sensitively detect AFB1 by real-time qPCR and the system was used in the real sample. Nevertheless, a few issues are needed to be clarified.

1. In section 2.5, the detail was missed about the solvent of AFB1, AFB2, AFG1, AFG2, OTA, or FB1. These toxins are not water – soluble, thus the information is important.

2.  Resolution of various figures is not high and can be improved. For example, figure 1B is fuzzy.

3. Please check the formatting of the manuscript. There are several cases when subscripted/superscripted text and font format do not appear correctly.

4. The title of 3.3 should be optimized. And the sentence in line 284-285 is confusing.

5. It is better to alter figure 6C into a table.

Author Response

1. In section 2.5, the detail was missed about the solvent of AFB1, AFB2, AFG1, AFG2, OTA, or FB1. These toxins are not water–soluble, thus the information is important.

Response: Yes, these toxins are not water-soluble. We dissolved these toxins in DMSO and then added them to the buffer. We have added the statement in section 2.5.

2. Resolution of various figures is not high and can be improved. For example, figure 1B is fuzzy.

Response: Figure 1B has improved the resolution quality.

3. Please check the formatting of the manuscript. There are several cases when subscripted/superscripted text and font format do not appear correctly.

Response: Checked and corrected.

4. The title of 3.3 should be optimized. And the sentence in line 284-285 is confusing.

Response: The title of 3.3 was changed to “The Specificity Test of the Selected Aptamer fl-2CS1”. The sentence was changed to “The specificity of the aptamers against their targets mostly relies on the tertiary structure of the aptamers.”

5. It is better to alter figure 6C into a table.

Response: Yes, we have done as suggested.

Reviewer 2 Report

grammar:

Aptamer-based sensors (aptasensors) has been should be "have been "

 Recently, the aptamers were used to detect AFB1 should be Recently, some aptamers were used to detect AFB1

The insert-containing plasmids were sequenced. It needs adding the company that did the analysis and the analysis methodology. 

The conversion result was examined on a TLC plate. TLC is a good method for the laboratory, but for the manuscript, either IR or NMR should be presented. 

I did not have access to the figures on the supplementary material.

it is confusing to the reader the sections saying flanking regions of the known sequence, if they are the primers, add them, even if this is in a supplementary table

in table 1, at the foot, please use same as Kd , instead of kd, also, if it has not been tested, should it be there?

in table 1, please explain more about Total number of the same sequence selected. is the number of clones?

the section of the mutants is really interesting and covers the use of a random aptamer or any piece of DNA oligo

at the beginning of the manuscript the authors indicate there is a need for aflatoxin sensors that could be cheaper, and more efficient, nevertheless, the authors refer to the use of PCR for detection, but this is not in line on the aspect of needing something cheaper or for easy access, could you please expand as to demonstrate using PCR is cheaper or more efficient that other available methodologies?

I am very happy to see the use of ITC in the study of aptamer - target, this is not something seen very often, but it is a great methodology to use and the authors should be pleased on doing a great job about it

Author Response

1. Aptamer-based sensors (aptasensors) has been should be "have been "

Response: changed.

2. Recently, the aptamers were used to detect AFB1 should be Recently, some aptamers were used to detect AFB1

Response: changed.

3. The insert-containing plasmids were sequenced. It needs adding the company that did the analysis and the analysis methodology. 

Response: The sentence was changed to “The insert-containing plasmids were sequenced with an ABI 3730XL DNA Analyzer served by G-TeC Genomic Technology Core (Academia Sinica, Taiwan).”

4. The conversion result was examined on a TLC plate. TLC is a good method for the laboratory, but for the manuscript, either IR or NMR should be presented. 

Response: Yes, IR or NMR is a good technique for the conversion result in detail. However, to examine the conversion efficiency as the case in this manuscript, TLC was quick and reliable for the next step of the experiment.

5. I did not have access to the figures on the supplementary material.

Response: The supplementary has been attached.

6. it is confusing to the reader the sections saying flanking regions of the known sequence, if they are the primers, add them, even if this is in a supplementary table

Response: Yes, the flanking sequences are the primer targeting sites. To clarify the statement, we rewrote the sentence as “The ssDNA pool consists of a 30-nt central randomized sequence flanked with 13 and 14 nt of the known sequence (the forward and reverse primer annealing site) at its 5'- and 3'-end, respectively. The sequence is shown in section 2.1.

7. in table 1, at the foot, please use same as Kd, instead of kd, also, if it has not been tested, should it be there?

Response: Changed.

8. in table 1, please explain more about Total number of the same sequence selected. is the number of clones?

Response: Yes, the “numbers” was the number of the same sequence selected. At the foot, we have changed it to “The total number of clones with the same sequence was selected.”

9. the section of the mutants is really interesting and covers the use of a random aptamer or any piece of DNA oligo

Response: After one round of counter-selection against NH2-MNPs, we sequenced ten clones, and four were in the same sequence. The known flanking sequences used for the primer binding involved in the structural formation that responds to bind the AFB1 were unexpected.

10. at the beginning of the manuscript the authors indicate there is a need for aflatoxin sensors that could be cheaper, and more efficient, nevertheless, the authors refer to the use of PCR for detection, but this is not in line on the aspect of needing something cheaper or for easy access, could you please expand as to demonstrate using PCR is cheaper or more efficient that other available methodologies?

Response: Yes, the hand-held device for real-time PCR was available. We have cited the reference Ahrberg et al., 2016 (reference no. 35 in the manuscript). Based on the technique of real-time PCR in this study, the detection ranges of 50 ppt to 50 ppb (Table 2) was sensitive enough to fit for the food safety limit (2 to 20 ppb). Besides, the company in Taiwan GeneReach Biotechnology (http://www.genereach.com/index.php) has a POCKIT-micro with a hand-held size. Due to the Covid pandemic, the expense of real-time PCR detection declined dramatically as the need for global uses. Therefore, we chose the real-time PCR for the detection could be cheaper than those mentioned before.

11. I am very happy to see the use of ITC in the study of aptamer - target, this is not something seen very often, but it is a great methodology to use and the authors should be pleased on doing a great job about it

Response: Yes, the ITC is a powerful tool in studying the interaction of aptamer against the target. It could provide the value of dissociation constant, ΔH, and ΔS. It could also test the specificity of the aptamer with the targets.

Round 2

Reviewer 1 Report

OK